

# A new sports garment with elastomeric technology optimizes physiological, mechanical, and psychological acute responses to pushing upper-limb resistance exercises

Angel Saez-Berlanga[1], Carlos Babiloni-Lopez[1], Ana Ferri-Caruana[1], Pablo Jiménez-Martínez[1,2], Amador García-Ramos[3,4], Jorge Flandez[5], Javier Gene-Morales[1] and Juan C. Colado[1]

[1] Research Group in Prevention and Health in Exercise and Sport (PHES), Department of Physical Education and Sports, University of Valencia, Valencia, Spain
[2] ICEN Institue, Madrid, Spain
[3] Department of Physical Education and Sport, Faculty of Sport Sciences, University of Granada, Granada, Spain
[4] Department of Sports Sciences and Physical Conditioning, Faculty of Education, Universidad Católica de la Santísima Concepción, Concepción, Chile
[5] Institute of Education Sciences, Austral University of Chile, Ciudad de Valdivia, Chile

Corresponding author
Javier Gene-Morales,
javier.gene@uv.es

## ABSTRACT

This study aimed to compare the mechanical (lifting velocity and maximum number of repetitions), physiological (muscular activation, lactate, heart rate, and blood pressure), and psychological (rating of perceived exertion) responses to upper-body pushing exercises performed wearing a sports elastomeric garment or a placebo garment. Nineteen physically active young adults randomly completed two training sessions that differed only in the sports garment used (elastomeric technology or placebo). In each session, subjects performed one set of seated shoulder presses and another set of push-ups until muscular failure. The dependent variables were measured immediately after finishing the set of each exercise. Compared to the placebo garment, the elastomeric garment allowed participants to obtain greater muscular activation in the pectoralis major (push-ups: $p = 0.04$, $d = 0.49$; seated shoulder press: $p < 0.01$, $d = 0.64$), triceps brachialis (push-ups, $p < 0.01$, $d = 0.77$; seated shoulder press: $p < 0.01$, $d = 0.65$), and anterior deltoid (push-ups: $p < 0.01$, $d = 0.72$; seated shoulder press: $p < 0.01$, $d = 0.83$) muscles. Similarly, participants performed more repetitions (push-ups: $p < 0.01$; $d = 0.94$; seated shoulder press: $p = 0.03$, $d = 0.23$), with higher movement velocity (all $p \leq 0.04$, all $d \geq 0.47$), and lower perceived exertion in the first repetition (push-ups: $p < 0.01$, $d = 0.61$; seated shoulder press: $p = 0.05$; $d = 0.76$) wearing the elastomeric garment compared to placebo. There were no between-garment differences in most cardiovascular variables (all $p \geq 0.10$). Higher diastolic blood pressure was only found after the seated shoulder press wearing the elastomeric garment compared to the placebo ($p = 0.04$; $d = 0.49$). Finally, significantly lower blood lactate levels were achieved in the push-ups performed wearing the elastomeric garment ($p < 0.01$; $d = 0.91$), but no significant differences were observed in the seated shoulder press ($p = 0.08$). Overall, the findings of this study suggest that elastomeric technology integrated into a sports

garment provides an ergogenic effect on mechanical, physiological, and psychological variables during the execution of pushing upper-limb resistance exercises.

## INTRODUCTION

Resistance training has several health and performance benefits, such as cardiovascular, body composition, biochemical, and functional improvements (*Suchomel et al., 2018*; *Fritz et al., 2018*). To maximize these benefits, training variables (*i.e.*, external and internal load parameters, exercise selection, and materials used) must be carefully manipulated (*Garber et al., 2011*; *Halson, 2014*). Accordingly, new tools or devices that can optimize training stimuli may be considered in the field of sports. Within this context, there is a lack of literature analyzing the external and internal load responses to exercises performed while wearing sports garments that incorporate elastomeric technology.

Training load (internal and external) can be evaluated in different ways (*Halson, 2014*). The main factor that provokes adaptations to resistance exercise is skeletal muscle contraction, which is controlled by the nervous system (*Alix-Fages et al., 2022*). Therefore, measuring neuromuscular responses to exercise provides relevant information regarding specific resistance training methodologies (*e.g.*, new training garments). Neuromuscular strategies during muscle contraction can be assessed using non-invasive surface electromyography (EMG) (*Hermens et al., 2000*). Mechanical performance (*e.g.*, number of repetitions completed, movement velocity, and kilograms lifted) provides further information on the specific mechanical responses to each exercise (*González-Badillo et al., 2017*). Alternatively, parameters of the internal load, such as metabolic responses to exercise, may be evaluated through the cardiovascular system (*e.g.*, heart rate (HR) and blood pressure) and metabolites such as blood lactate (*Wirtz et al., 2014*). In this regard, lactate and heart rate are commonly used to quantify training intensity, as they are positively correlated with training intensity (*Beneke, Leithäuser & Ochentel, 2011*; *Mann, Lamberts & Lambert, 2013*). Finally, the rating of perceived exertion (RPE) is correlated with different exercise outcomes (*e.g.*, weight used and HR) (*Morishita et al., 2019*). A wide range of subjective scales has been validated to verify exercise intensity with different training devices (*e.g.*, elastic bands (EB) or weight plates) and populations (*e.g.*, trained youth and older adults) (*Colado et al., 2012*, *2014*, *2018*, *2020a*, *2023*).

These psychophysiological and/or biomechanical outcomes may be optimized using different training methods and/or tools (*Andersen et al., 2020*; *Babiloni-Lopez et al., 2022*). Historically, athletes have been instructed to lift external resistances as fast as possible to maximize adaptations in their rate of force development (RFD). A disadvantage of this instruction is that a large portion of the range of motion is spent decelerating the resistance (*Rhea, Kenn & Dermody, 2009*). Different devices have appeared to overcome this

limitation in the last decades, and scientists have analyzed their potential positive effects (*Youdas et al., 2010*; *Porcari et al., 2011*; *Parry, Straub & Cipriani, 2012*; *Borreani et al., 2015*; *Calatayud et al., 2015b*). In this regard, elastic devices can enhance physical capabilities because of their verified results, inexpensive acquisition, and easy portability (*Colado et al., 2020b*; *Saez-Berlanga et al., 2022*; *Babiloni-Lopez et al., 2022*). Furthermore, the use of variable resistance provides benefits in hypertrophy, strength, and power (*Suchomel et al., 2018*). Variable resistance is commonly incorporated while performing free-weight exercises (*Suchomel et al., 2018*).

Resistance training with elastic bands allows exercisers to employ a mechanical advantage to produce both high-power and high-force levels (*Gaamouri et al., 2023*). Considering the elongation coefficient, the bands provide more resistance when stretched and less when shortened (*Gene-Morales et al., 2020*). Therefore, depending on how they are applied (*i.e.*, in the same direction as the concentric or eccentric phase), elastic bands can assist or resist movements. Participants can achieve higher muscle torque production and muscle activation levels when elastic bands resist movement (*Aboodarda et al., 2013*). In contrast, the elastic bands used for assistance allow exercisers to lift greater loads at higher velocities (*Andersen et al., 2019*). Another key factor of elastic bands is that they must be attached to a structure and bar to be lifted (*Gene-Morales et al., 2020*) or directly grabbed with the hands (*Treiber et al., 1998*; *Page et al., 2015*). This could condition the technique of certain movements and limits to work on-site and/or other constraints. Consequently, new devices, such as sports garments incorporating elastomeric technology, may overcome the limitations of traditional elastic bands. Specifically, a new sports garment that incorporates front and back elastomers around the chest and along the arms could allow for free movement and, depending on joint positioning, could assist or resist movement. For instance, during the last degrees of the eccentric phase in pushing exercises, when the chest is opened and the scapulae are retracted, the front elastomers are stretched and assist the movement. Conversely, the back elastomers are slack and, therefore, do not provide any resistance. Subsequently, with the progression of the concentric phase, the front elastomers are shortened and do not provide resistance, and the elastomers from the back begin to stretch and provide resistance. Bearing this in mind, the question arises as to whether this new sports garment incorporating elastomeric technology can further potentiate the benefits previously described for elastic variable resistance training.

Therefore, this study aimed to compare the mechanical performance (lifting velocity and maximum number of repetitions), physiological (neuromuscular activity, blood lactate, heart rate, and blood pressure), and psychological (rating of perceived exertion) responses during pushing exercises (seated shoulder press and push-up) performed wearing or not wearing a new sports garment for the upper body that incorporates elastomeric technology. Considering elastomeric properties, we hypothesized that the use of the elastomeric garment would allow for the execution of more repetitions, greater lifting velocities, and greater neuromuscular activity, while no significant differences were expected for blood lactate concentration, cardiovascular responses, and perceived effort.

## MATERIALS AND METHODS

### Participants

The inclusion criteria were: (a) young adults between 18 and 30 years old with a minimum of 1 year of resistance training experience, and (b) the participants had to be free from cardiovascular or osteoarticular disease history, or clinical, neuromotor, or cognitive contraindications for the performance of the physical tests. Finally, 19 physically active men were selected through convenience sampling and voluntarily participated in this study. Participants were instructed not to eat, take stimulants (*e.g.*, caffeine), or other ergogenic substances 3 to 4 h before the sessions, and not to perform intense physical activity or exercise for the upper limbs 24 h before the study. They were encouraged to sleep for at least 8 h the night before data collection.

The participants were carefully informed about the potential risks and discomfort of the project and signed a written consent form before the start of the study. The study protocol was approved by the Human Research Ethics Committee of the University of Valencia (H20190325095509) and conducted in accordance with the Declaration of Helsinki.

### Procedures

A randomized, crossover, within-participant study design was used to explore the effects of performing seated shoulder presses at 70% of one repetition maximum (1RM) and push-ups while wearing a new sports garment for the upper body that incorporates elastomeric technology. The study was conducted over 8 weeks at the Faculty of Physical Activity and Sports Sciences of the University of Valencia (Spain). Each participant completed three sessions separated by 48 h: (a) one for familiarization and preliminary assessments, and (b) two experimental sessions. A subsample extracted from the general sample participated in an additional session to assess intersession reliability. This additional session was separated from the last experimental session by 72 h. The overall study design is shown in Fig. 1. All sessions lasted approximately 60 min and were conducted between 10:00 and 13:00 h to avoid circadian variations in the performance of the dependent variables (*Sundstrup et al., 2012*). Each subject performed both experimental sessions within the same hour. All measurements were conducted by the same investigators and always performed in the same sports facility. A minimum ratio of 4:1 was maintained between the researchers and participants.

#### Familiarization

The familiarization session was used to (i) characterize the participants through an interview and anthropometric measurements, (ii) teach the participants the specific standardized protocol and technique of the exercises (*Colado & García-Massó, 2009*) using the two sports garments (*i.e.*, with elastomeric technology and without it, placebo); (iii) report the RPE for active muscles at the first and last repetition of each set (*Colado et al., 2023*); and (iv) estimate the 70% 1RM load for the seated shoulder press.

After a brief interview, body weight and fat percentage were measured using an electrical bioimpedance device (Tanita BF-350, Tanita Corp., Tokyo, Japan). Body height was determined to the nearest 0.5 cm during maximum inhalation using a wall stadiometer

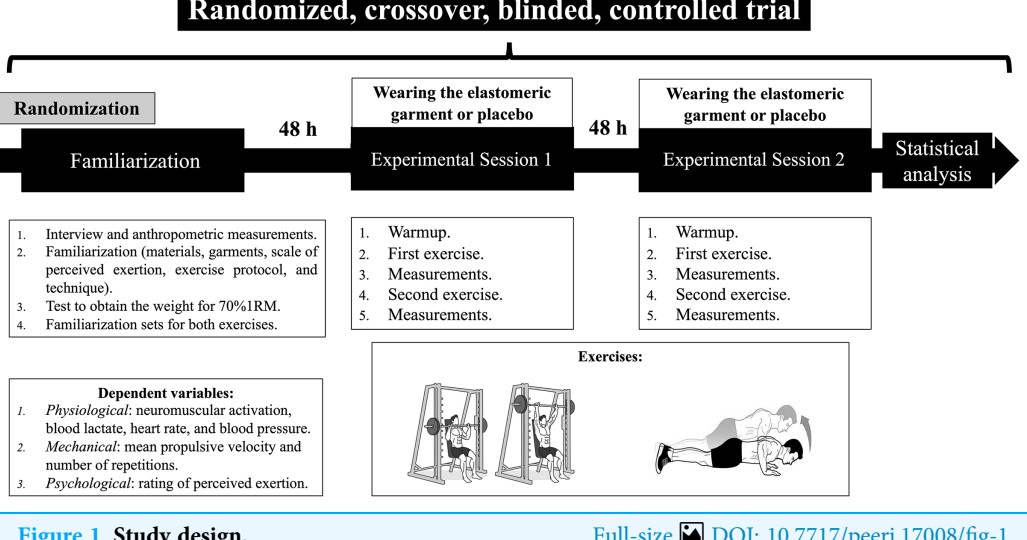

**Figure 1** Study design.

(Seca T214; Seca Ltd., Hamburg, Germany). Handgrip strength was assessed using a dynamometer (Scacam-EH10117; Scacam®, South El Monte, California, United States of America) based on the protocol of a previous study (*Leong et al., 2015*): (i) the participant stood; (ii) with the elbow extended, and the shoulder, forearm, and wrist in a neutral position; and (iii) the participant was required to exert maximal grip force for 5 s. Two attempts were performed with 1 min of rest in between.

At this point, the participants were shown the devices that were used throughout the study. A Smith machine (Multipower Powerline PSM144X; Powerline, Forest Park, Illinois, United States) was used to perform the seated shoulder press. For the RPE, the OMNI-Resistance Exercise Scale for elastic bands was chosen (*Colado et al., 2018*), which was visible to the participants at every moment during the execution of the exercises. Finally, participants were shown both garments to use in the study: the sports garment with elastomeric technology (Pro-Advance; Menatechpro System®, Madrid, Spain) (see Supplemental Figures) and an equivalent sports garment used as placebo (same garment but not including elastomers). Menatechpro System® elastomeric technology is a patented, sophisticated sportswear that generates elastic resistance in most planes of motion through the elongation of the elastomers included in the garment. More specifically, the elastomeric garment includes front and back elastomers around the chest. These elastomers connect on the shoulders and descend to the hand through each arm with two elastomer lines. According to the disposition of the elastomers, the garment may also assist in certain parts of the movement and function in a certain manner as a compression garment. This garment is composed of more than 20 pieces. Specifically, the model used in the present study (Pro-Advance) provides a resistance of eight kilograms at the maximum elongation. This sports garment is recommended for users with previous training experience, who want to enhance both their physical performance and the intensity of their resistance training. Twelve similar garments (six garments with elastomeric technology and six placebo garments) were employed during the study to

better adapt the garment to the anthropometric characteristics of each participant. Fair to excellent intersession relative reliability, with intraclass correlation coefficients (ICC) between 0.60 and 0.99 for the conditions performed wearing the elastomeric garment and between 0.57 and 0.99 for the conditions performed with the placebo garment were obtained. Similarly, good-to-excellent intersession absolute reliability was observed, with coefficients of variation between 2.54% and 24.33% for the conditions performed wearing the elastomeric garment and between 2.29% and 21.91% for the conditions performed with the placebo garment. The specific relative and absolute intersession reliability values for each dependent variable are shown in Tables S1 and S2.

Subsequently, the participants were instructed on the standardized warm-up, which consisted of dynamic stretching and isometric exercises, as in previous studies (*Calatayud et al., 2014*; *Gene-Morales et al., 2023*). Movement intensities were gradually increased during warm-up to prepare participants for peak performance during the strength test. Subsequently, the technique of both exercises was explained. To homogenize the speed of movement, the execution tempo for both exercises consisted of a maximum-speed concentric phase and a three-second eccentric phase, with no pauses. The tempo of the eccentric phase was controlled using a metronome (Ableton Live 6; Ableton AG, Berlin, Germany) at 60 beats per minute. Additionally, the participants received constant verbal and visual feedback to maintain a proper body position and range of motion. The following techniques and positions were adopted for the seated shoulder press: (i) upright seated position with back support, (ii) hips and knees at 90°flexion, (iii) feet on the floor at hip width, (iv) elbows and shoulders flexed to align the bar with the chin, and (v) standardized biacromial grip width. From this starting position (bar aligned with the chin), the participant lifted the bar to a position with outstretched elbows. For the push-ups, according to a previous study (*Calatayud et al., 2015a*), each participant (i) started in an outstretched arms position, (ii) with fingers slightly abducted and extended, (iii) with neutral spine and hips, and (iv) feet at hip width. By flexing their elbows, the participants had to lower until their chest was aligned with their hand. Maintaining a neutral spine and neutral hips was mandatory throughout the set.

Finally, we estimated the weight to be used for the seated shoulder press corresponding to 70% 1RM of each participant. This estimation was based on the %1RM-velocity profile (*García-Ramos, Suzovic & Pérez-Castilla, 2021*). More specifically, the participants performed one repetition with an agreed weight of 70% of 1RM. If the MPV fell between 0.61 and 0.69 meters per second (*García-Ramos, Suzovic & Pérez-Castilla, 2021*), this was the weight to use. If the MPV was outside this range, the weight was modified and another attempt was performed after 5 min of passive rest. Once the appropriate weight was obtained, a second repetition with the same load was performed, after a 5-min rest, to ensure reliability. To end with the familiarization session, participants performed one set to failure of each exercise (at 70% 1RM for the seated shoulder press and bodyweight for the pushups), reporting the RPE at the first and last repetition of each set. A 5-min rest period was allowed for these sets.

**Table 1 Description of the physiological variables.**

| Variable | Abbreviation | Description |
|---|---|---|
| Pectoralis major neuromuscular activation (µV) | RMSPEC | Root mean square (RMS) of the electromyographic activity of the clavicular fibers of the pectoralis major |
| Triceps brachii neuromuscular activation (µV) | RMSTRI | Root mean square (RMS) of the electromyographic activity of the long head of the triceps brachii |
| Anterior deltoid neuromuscular activation (µV) | RMSDELT | Root mean square (RMS) of the electromyographic activity of the anterior deltoid |
| Rectus abdominis neuromuscular activation (µV) | RMSABD | Root mean square (RMS) of the electromyographic activity of the upper rectus abdominis |
| Blood lactate (mmol/L) | BL | Metabolite mainly produced in the skeletal muscle that serves as an indicator of exercise intensity (*Foucher & Tubben, 2023*) |
| Heart rate (bpm) | HR | Number of heartbeats for 1 min |
| Systolic blood pressure (mmHg) | SBP | Peak arterial pressure when ventricles pump blood out of the heart (*Rehman, Hashmi & Nelson, 2022*) |
| Diastolic blood pressure (mmHg) | DBP | Minimum pressure between beats when the heart is filling with blood (*Rehman, Hashmi & Nelson, 2022*) |

### Experimental sessions

The study consisted of two experimental sessions, one to be fully performed while wearing the elastomeric garment and the other with the placebo garment. The order of the experimental sessions was randomized (https://random.org/lists) for each participant. Push-ups and seated shoulder presses were included in both experimental sessions. Therefore, four conditions were performed: (a) push-ups wearing the elastomeric garment, (b) seated shoulder press wearing the elastomeric garment, (c) push-ups wearing the placebo garment, and (d) seated shoulder press wearing the placebo garment. The order in which the exercises were performed was randomized in the first experimental session and maintained for the second experimental session for each participant. Upon arrival at the laboratory in both experimental sessions, the participants rested seated for 10 min while listening to self-selected music (*Greco et al., 2022*) to induce similar inter-session resting homeostatic conditions. The dependent variables, including the physiological, mechanical, and psychological variables, are outlined in Tables 1–3.

In the first experimental session, after the warm-up, the participants performed seated shoulder presses and push-ups in the pertinent order. A 10-min rest was allowed between the exercises. The dependent variables were measured at this point. More specifically, the mechanical variables (lifting velocity and maximum number of repetitions) were recorded during the performance of the exercise, as was the RPE (psychological variable), which was verbalized by each participant at the end of the first and last repetitions. The physiological variables were measured immediately after each exercise. Measurements were taken with the participant seated in an adjacent space separated from the exercise area by a partition screen to blind the researcher in charge. The same procedure was followed in the second experimental session.

**Table 2 Description of the mechanical variables.**

| Variable | Abbreviation | Description |
|---|---|---|
| Mean propulsive velocity of the first repetition (m/s) | 1stMPV | The average velocity achieved during the acceleration phase (*García-Ramos et al., 2018*) of the first repetition |
| Mean propulsive velocity of the last repetition (m/s) | LMPV | The average velocity achieved during the acceleration phase (*García-Ramos et al., 2018*) of the last repetition |
| Mean propulsive velocity peak (m/s) | PMPV | Highest mean propulsive velocity of the set |
| Mean propulsive velocity average (m/s) | AMPV | The average mean propulsive velocity of the set |
| Number of repetitions | Rep | Total number of valid repetitions executed |

**Table 3 Description of the psychological variables.**

| Variable | Abbreviation | Description |
|---|---|---|
| Rating of perceived exertion of the first repetition | 1stRPE | Perceived effort (between 0 and 10) of the first repetition |
| Rating of perceived exertion of the last repetition | LRPE | Perceived effort (between 0 and 10) of the last repetition |

## Measurement equipment and data acquisition
### Physiological variables: neuromuscular activation

The EMG signal was obtained using two two-channel handheld devices (Realtime Technologies Ltd., Dublin, Ireland) with 16-bit analog-to-digital (A/D) conversion. EMG data were monitored using validated mDurance software for Android (mDurance Solutions S.L., Granada, Spain). Surface Electromyography for the Non-Invasive Assessment of Muscles criteria (SENIAM) (*Hermens et al., 2000*) and previous studies in this field (*Calatayud et al., 2017*) were followed.

To ensure consistency in electrode placement, each participant was shaved and cleaned with a cotton swab moistened with alcohol (*Calatayud et al., 2014*). Surface electrodes were placed on the anterior deltoid, clavicular fibers of the pectoralis major, upper rectus abdominis, and the long head of the triceps brachii. Chlorinated silver pre-gelled bipolar surface electrodes (Kendall™ Medi-Trace, Coividien, Barcelona, Spain) were placed at an inter-electrode distance of 10 mm. The reference electrode was placed over the nearest bone prominence (in our study, the acromion and the superior iliac spine). A mark was made on the skin of the participants around each electrode with a permanent marker to easily place the electrodes in the next session and to ensure reliability. One device collected EMG data from the anterior deltoid and clavicular bundles of the pectoralis major muscles, while the other collected data from the upper rectus abdominis and long head of the triceps brachii muscles. The sampling rate was planned at 1,024 Hz.

Data were collected as described by *Ferri-Caruana et al. (2022)* and *Gene-Morales et al. (2023)*. Specifically, all the EMG signals were stored on a hard disk for subsequent evaluation. mDurance software digitally filtered the raw signals automatically using a fourth-order *"Butterworth"* bandpass filter between 20 and 450 Hz. A high-pass cut-off frequency of 20 Hz was employed to reduce any *"artifacts"* that might occur throughout

the movement to have a minimum impact on the total power recorded by the EMG (*Ferri-Caruana et al., 2022*). Before carrying out the tests, the participants performed one repetition of the seated shoulder press and another of push-ups to check for proper signal saturation. Finally, the average EMG signals (measured by the root-mean-square (RMS)) of all the effective repetitions performed in each set were retained for analysis.

### Physiological variables: blood lactate

BL concentrations were measured from capillary blood extracted from the fingertips. Blood samples were collected before the session and immediately after each exercise and were analyzed using a portable lactate analyzer (Lactate Pro 2; Arkray Inc., Kyoto, Japan).

### Physiological variables: cardiovascular parameters

Pre- and post-test heart rate (HR), systolic (SBP), and diastolic blood pressure (DBP) were monitored *via* digital wrist blood pressure monitor (RS4-model; Omron Electronics Iberia SAU, Valencia, Spain).

### Mechanical variables: mean propulsive velocity and number of repetitions

A linear position transducer (ADR Encoder, ADR, Toledo, Spain) was used to collect the MPV (m/s). More specifically, we analyzed the (i) MPV of the first repetition (1$^{st}$MPV) (ii) and last repetition (LMPV), (iii) peak MPV of the set (PMPV), (iv) average MPV of the set (AMPV), and (v) the number of repetitions. For the seated shoulder press, the transducer was attached to the bar, allowing it to be moved vertically at maximum velocity (*Naclerio et al., 2011*). For the push-ups, each participant wore a strap around their chest at the level of the xiphoid process. The transducer was attached to this strap.

### Psychological variables: rating of perceived exertion

Participants reported the RPE for the active muscles at the end of the first (1$^{st}$RPE) and last repetition (LRPE). The OMNI-RES Scale for Elastic Bands (*Colado et al., 2012*) was always used regardless of the condition analyzed. Previous research demonstrated that the 1$^{st}$RPE could be used independently of the age, sex, or fitness level of the participants (*Pincivero, Timmons & Elsing, 2010*; *Babiloni-Lopez et al., 2022*; *Colado et al., 2023*)

## Statistical analyses

Statistical analyses were performed using commercial software (SPSS version 28.0; IBM Corp., Armonk, New York, USA). The assumption of normality of the dependent variables was verified using the Shapiro-Wilk test. Almost all the variables showed a normal Gaussian distribution. The variables showing a nonnormal distribution were the electromyographic activity of the triceps brachii (RMSTRI), 1$^{st}$MPV, number of repetitions, and RPE of both exercises; the electromyographic activity of the anterior deltoid (RMSIDELT) and rectus abdominis (RMSABD) of the seated shoulder press; and the SBP, and DBP of the push-ups. Results are reported as the mean ± standard deviation (SD). The level of statistical significance was set at $p \leq 0.05$.

Parametric two-tailed Student's t-test of related samples or nonparametric Wilcoxon test assessed the differences between performing each exercise wearing the elastomeric garment or the placebo garment. The effect size was calculated by means of Cohen's d,

which was interpreted as a low (<0.50), moderate (0.50–0.79), or large effect (≥0.80) (*Cohen, 1988*).

The test-retest relative reliability was assessed using the intraclass correlation coefficient (ICC, model 3.1) (*Yen & Lo, 2002*). As previously suggested, ICC values were interpreted as poor (<0.50), moderate (0.50–0.75), good (0.75–0.90), and excellent (>0.90) reliability (*Koo & Li, 2016*). On the other hand, the absolute reliability was evaluated as previously described (*Hopkins, 2000*; *Gene-Morales et al., 2022*). More specifically, we used the coefficient of variation (CV = (standard error of measurement/mean of both measurements) × 100; where the standard error of measurement is the standard deviation of the difference between the two measurements divided by the square root of the number of measurements per subject) (*Hopkins, 2000*). We defined excellent absolute reliability as CV ≤ 10%, good with a CV between 10–20%, acceptable with a CV between 20–30%, and poor as CV > 30% (*Aronhime et al., 2014*). Data for the reliability calculations were obtained from an additional session in which a subsample of participants performed for a second time the same exercises under the same conditions (*i.e.*, push-ups wearing the elastomeric garment, seated shoulder press wearing the elastomeric garment, push-ups wearing the placebo garment, and seated shoulder press wearing the placebo garment).

# RESULTS

## Participants

The sample size was determined using G* Power 3.1 software (*Faul et al., 2009*) based on previous pilot studies (*Gene-Morales et al., 2023*). This a-priori analysis was performed to reduce the probability of type II error and determine the minimum number of participants required to reject the null hypothesis at the $p < 0.05$ level of confidence (*Beck, 2013*). The calculation indicated that 18 volunteers were necessary to meet the required power of 0.90, α of 0.05, and effect size dz of 0.82. Finally, a total of 19 healthy, trained subjects were included. None of the participants dropped out of the study. Descriptive data of the participants in this study were: age = 24.7 ± 4.9 years; height = 178.8 ± 4.5 cm; body mass = 78.1 ± 9.0 kg; body fat percentage = 14.1 ± 4.1%; manual dynamometry = 48.2 ± 8.5 kg, resistance training experience = 2.8 ± 1.8 years; weekly training frequency = 3.9 ± 1.0 days/week.

## Physiological variables

Descriptive and inferential analyses of the physiological outcomes included in the study are presented in Table 4. Furthermore, Fig. 2 shows the graphical representation of the EMG results.

### Neuromuscular activation

Wearing the elastomeric garment to perform the seated shoulder press and push-ups entailed greater neuromuscular activation in the pectoralis major (RMSPEC, push-ups: $p = 0.04$, $d = 0.49$; seated shoulder press: $p < 0.01$, $d = 0.64$), triceps brachialis (RMSTRI, push-ups: $p < 0.01$, $d = 0.77$; seated shoulder press: $p < 0.01$, $d = 0.65$), and anterior deltoid (RMSDELT, push-ups: $p < 0.01$, $d = 0.72$; seated shoulder press: $p < 0.01$, $d = 0.83$)

**Table 4 Mechanical performance outcomes to both exercises performed wearing the elastomeric garment or the placebo.** Significant difference ($p < 0.05$). Results are presented as mean ± standard deviation, mean difference (m.d.), 95% confidence interval between brackets, significance (p), and effect size measured through Cohens'$d$ (interpreted as low (<0.50), moderate (0.50–0.79), or large (≥0.80)). 1$^{st}$MPV, mean propulsive velocity of the first repetition; LMPV, mean propulsive velocity of the last repetition; PMVP, peak mean propulsive velocity of the set; AMVP, average mean propulsive velocity of the set; m/s, meters per second.

| Variable | Push-ups | | | Seated shoulder press | | |
|---|---|---|---|---|---|---|
| | Placebo | Elastomeric garment | Paired differences | Placebo | Elastomeric garment | Paired differences |
| RMSPEC (µV) | 689.76 ± 198.98 | 777.45 ± 226.17 | m.d. = 87.68 (48.81–174.55) $p = 0.04^* d = 0.49$ | 569.69 ± 243.84 | 668.23 ± 307.77 | m.d. = 98.54 (24.54–172.53) $p = 0.01^* d = 0.64$ |
| RMSTRI (µV) | 246.09 ± 107.53 | 300.21 ± 137.73 | m.d. = 54.11 (20.13–88.10) $p < 0.01^* d = 0.77$ | 296.86 ± 118.27 | 392.36 ± 164.98 | m.d. = 95.50 (24.12–166.87) $p < 0.01^* d = 0.65$ |
| RMSDELT (µV) | 523.51 ± 166.03 | 614.78 ± 204.37 | m.d. = 91.27 (30.28–152.27) $p < 0.01^* d = 0.72$ | 903.10 ± 316.42 | 1053.84 ± 288.89 | m.d. = 150.73 (63.53–237.93) $p < 0.01^* d = 0.83$ |
| RMSABD (µV) | 68.68 ± 19.12 | 70.78 ± 16.75 | m.d. = 2.10 (−4.07 to 8.28) $p = 0.48$ | 70.63 ± 16.78 | 73.55 ± 32.97 | m.d. = 2.92 (−10.82 to 16.67) $p = 0.44$ |
| Blood Lactate (mmol/L) | 7.42 ± 1.60 | 6.61 ± 1.52 | m.d. = 0.81 (0.38–1.24) $p < 0.01^* d = 0.91$ | 5.01 ± 1.19 | 4.63 ± 1.25 | m.d. = 0.37 (−0.05 to 0.80) $p = 0.08$ |
| Heart Rate (bpm) | 93.21 ± 18.86 | 88.47 ± 14.73 | m.d. = 4.73 (−2.70 to 12.18) $p = 0.19$ | 78.95 ± 11.73 | 82.47 ± 19.98 | m.d. = 3.52 (−3.98 to 11.04) $p = 0.33$ |
| SBP (mmHg) | 132.84 ± 16.48 | 132.21 ± 18.09 | m.d. = 0.63 (−8.83 to 10.10) $p = 0.98$ | 127.00 ± 14.47 | 133.84 ± 10.03 | m.d. = 6.84 (−1.51 to 15.20) $p = 0.10$ |
| DBP (mmHg) | 92.16 ± 16.87 | 93.53 ± 17.87 | m.d. = 1.36 (−6.49 to 9.23) $p = 0.88$ | 85.84 ± 11.20 | 93.58 ± 11.12 | m.d. = 7.73 (0.15–15.32) $p = 0.04^* d = 0.49$ |

**Note:**
* Significant difference ($p < 0.05$). Results are presented as mean ± standard deviation, mean difference (m.d.), 95% confidence interval between brackets, significance ($p$), and effect size measured through Cohens'$d$ (interpreted as low (<0.50), moderate (0.50–0.79), or large (≥0.80)). RMSPEC, root-mean-square of the pectoralis major activation; RMSTRI, root-mean-square of the triceps brachialis activation; RMSDELT, root-mean-square of the anterior deltoid activation; RMSABS, root-mean-square of the rectus abdominis activation; µV, microvolts; mmol/L, millimole per liter; bpm, beats per minute; mmHg, millimeters of mercury.

compared to the same exercises performed with the placebo garment. Nonsignificant differences were found in the RMSABD (push-ups: $p = 0.48$; seated shoulder press: $p = 0.44$).

### Blood lactate

Significantly ($p < 0.01$; $d = 0.91$) less blood lactate was accumulated after performing push-ups wearing the elastomeric garment compared to the placebo. Nonsignificant differences were found after the seated shoulder press ($p = 0.08$).

### Heart rate

Nonsignificant differences were found in the heart rate after any of the exercises (push-ups: $p = 0.19$; seated shoulder press: $p = 0.33$).

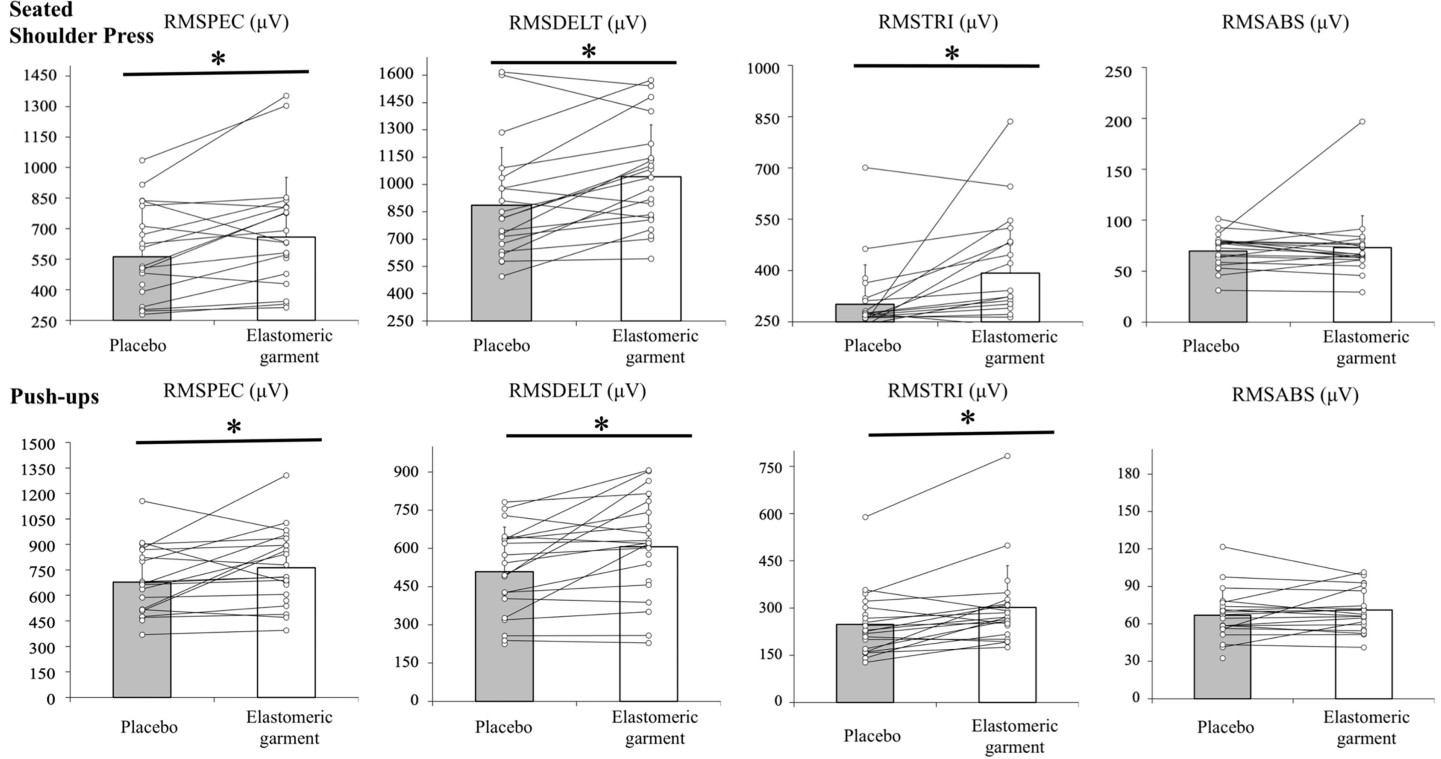

**Figure 2** **Interindividual variation of the neuromuscular activation when performing a seated shoulder press (top graphs) and push-ups (bottom graphs) wearing the elastomeric garment or placebo.** RMS, root mean square of the electromyographic values measured in µV (microvolts). An asterisk (*) indicates significant differences ($p < 0.05$) between the condition performed with the elastomeric garment and the placebo garment.

### Systolic and diastolic blood pressure

Participants achieved greater post-exercise DBP when performing the seated shoulder press with the elastomeric garment compared to the placebo ($p = 0.04$; $d = 0.49$). Nonsignificant effects were found in the SBP after any of the exercises (push-ups: $p = 0.98$; seated shoulder press: $p = 0.10$) or the DBP in the push-ups ($p = 0.88$).

## Mechanical variables

Table 5 presents the descriptive and inferential statistical comparisons of the mechanical performance.

### Mean propulsive velocity

The participants achieved greater 1st MPV ($p < 0.01$; $d = 1.14$), PMPV ($p < 0.01$; $d = 0.92$), and AMPV ($p < 0.01$; $d = 0.47$) performing the push-ups with the elastomeric garment compared to the placebo. In the shoulder press, participants obtained greater 1st MPV ($p < 0.01$; $d = 1.05$) and PMPV ($p = 0.04$; $d = 0.47$) wearing the elastomeric garment compared to the placebo.
**Table 5 Mechanical performance outcomes to both exercises performed wearing the elastomeric garment or the placebo.**

| Variable | Push-ups | | | Seated shoulder press | | |
|---|---|---|---|---|---|---|
| | Placebo | Elastomeric garment | Paired differences | Placebo | Elastomeric garment | Paired differences |
| 1stMPV (m/s) | 0.57 ± 0.08 | 0.63 ± 0.10 | m.d. = 0.06 (0.03–0.09) $p < 0.01^* d = 1.14$ | 0.54 ± 0.05 | 0.57 ± 0.06 | m.d. = 0.03 (0.01–0.04) $p < 0.01^* d = 1.05$ |
| LMPV (m/s) | 0.19 ± 0.03 | 0.18 ± 0.03 | m.d. = 0.01 (−0.01 to 0.03) $p = 0.30$ | 0.20 ± 0.04 | 0.20 ± 0.04 | m.d. = 0.00 (−0.01 to 0.03) $p = 0.57$ |
| PMPV (m/s) | 0.60 ± 0.09 | 0.65 ± 0.10 | m.d. = 0.05 (0.02–0.09) $p < 0.01^* d = 0.92$ | 0.55 ± 0.05 | 0.57 ± 0.06 | m.d. = 0.23 (0.00–0.04) $p = 0.04^* d = 0.47$ |
| AMPV (m/s) | 0.45 ± 0.07 | 0.48 ± 0.08 | m.d. = 0.03 (0.00–0.05) $p = 0.01^* d = 0.47$ | 0.40 ± 0.04 | 0.43 ± 0.06 | m.d. = 0.01 (0.00–0.04) $p = 0.12$ |
| Repetitions | 18.68 ± 4.48 | 21.37 ± 5.51 | m.d. = 2.68 (1.30–4.06) $p < 0.01^* d = 0.94$ | 9.16 ± 2.36 | 9.68 ± 1.70 | m.d. = 0.52 (−0.60 to 1.65) $p = 0.03^* d = 0.23$ |

Note:
* Significant difference ($p < 0.05$). Results are presented as mean ± standard deviation, mean difference (m.d.), 95% confidence interval between brackets, significance ($p$), and effect size measured through Cohens'$d$ (interpreted as low (<0.50), moderate (0.50–0.79), or large (≥0.80)). 1stMPV: mean propulsive velocity of the first repetition; LMPV: mean propulsive velocity of the last repetition; PMVP: peak mean propulsive velocity of the set; AMVP: average mean propulsive velocity of the set; m/s: meters per second.

**Table 6 Psychological responses to both exercises performed wearing the elastomeric garment or the placebo.**

| Variable | Push-ups | | | Seated shoulder press | | |
|---|---|---|---|---|---|---|
| | Placebo | Elastomeric garment | Paired differences | Placebo | Elastomeric garment | Paired differences |
| 1stRPE | 2.00 ± 0.66 | 1.53 ± 0.61 | m.d. = 0.47 (0.17–0.76) $p < 0.01^* d = 0.61$ | 3.16 ± 0.76 | 2.79 ± 0.78 | m.d. = 0.36 (0.00–0.73) $p = 0.05^* d = 0.76$ |
| LRPE | 9.58 ± 0.50 | 9.47 ± 0.51 | m.d. = 0.10 (−0.11 to 0.32) $p = 0.31$ | 9.68 ± 0.47 | 9.58 ± 0.50 | m.d. = 0.10 (−0.11 to 0.32) $p = 0.31$ |

Note:
* Significant difference ($p < 0.05$). Results are presented as mean ± standard deviation, mean difference (m.d.), 95% confidence interval between brackets, significance ($p$), and effect size measured through partial Cohens'$d$ (interpreted as low (<0.50), moderate (0.50–0.79), or large (≥0.80)). 1stRPE, rate of perceived exertion of the first repetition; LRPE, rate of perceived exertion of the last repetition.

### Number of repetitions

The use of the elastomeric garment allowed significantly more repetitions compared to the placebo in both exercises (push-ups: $p < 0.01$; $d = 0.94$; seated shoulder press: $p = 0.03$, $d = 0.23$).

## Psychological variables

Descriptive and inferential comparisons of the rate of perceived exertion outcomes are presented in Table 6.

*Rating of perceived exertion*

Both exercises performed wearing the elastomeric garment entailed lower $1^{st}$RPE (push-ups: $p < 0.01$, $d = 0.61$; seated shoulder press: $p = 0.05$; $d = 0.76$) compared to the same exercises performed with the placebo garment. Nonsignificant differences were encountered in the LRPE (push-ups: $p = 0.31$; seated shoulder press: $p = 0.31$).

## DISCUSSION

To the best of our knowledge, this is the first study analyzing the physiological, mechanical, and psychological responses to a seated shoulder press and push-ups performed wearing a new sports garment that incorporates elastomeric technology. The main finding was that performing both exercises wearing the elastomeric garment significantly improved physiological, mechanical, and psychological responses compared to the placebo garment. Although only one pilot study analyzed this specific garment (*Gene-Morales et al., 2023*), the results are consistent with the well-known positive effects of training with variable resistances such as elastic bands and chains (*Suchomel et al., 2018*; *Colado et al., 2020b*; *Hammami et al., 2022*).

### Physiological variables

Regarding muscular activation, the elastomeric garment allowed participants to obtain higher muscular activation on all muscles, except the rectus abdominis. This is probably due to the additional eight kilograms provided by the elastomeric garment at maximum elbow extension, being the weight a main factor that conditions muscular activation (*Schoenfeld et al., 2014*). The nonsignificant differences reported in the rectus abdominis may be due to the elastomeric garment not resisting the trunk movements. Apart from the additional load provided by the elastomeric garment, it is worth considering the elongation coefficient (*Andersen et al., 2020*). The elastomers display the resistance progressively throughout the range of motion, providing less load at the "sticking region" (see *Kompf & Arandjelović, 2016* for further information), and greater resistance during the biomechanically advantageous phase after the sticking point (*Iversen et al., 2017*). As a result, the elastomeric garment could help to overcome the sticking region and, therefore, optimize the neuromuscular response to resistance exercise (*Kompf & Arandjelović, 2016*). Another factor that could facilitate greater neuromuscular activation is the overload generated by the elastomeric garment during the first degrees of the eccentric phase, which, although not measured in this study, could help to increase the stimulus. This has been proven by previous research, which found that the use of elastic bands increases the resistance used in the eccentric phase and does not modify the technique or the neuromuscular performance during the concentric response (*Aboodarda et al., 2014*).

Besides the positive results obtained in terms of muscle activation, a significantly reduced blood lactate concentration was observed after the push-ups performed with the elastomeric garment compared to the placebo garment. This can be attributed to the fact that wearing compressive garments can increase venous blood flow (*Liu et al., 2008*). In this sense, the compression of the superficial tissues of the extremities reduces the diameter of the underlying veins, speeding the blood flow and improving the venous return

(*Liu et al., 2008*). Another reason for the increased blood flow and post-exercise lactate reduction may be linked to venular-arteriolar communication. External compression can reduce the venular lumen and increase shear stress, which would trigger the release of endothelial dilators (*Paszkowiak & Dardik, 2003*) and cause an eventual dilation in neighboring arteries (*Bochmann et al., 2005*). The nonsignificant differences observed in the seated shoulder press could be due to the shorter duration (approximately nine repetitions for the shoulder press compared to approximately 20 push-ups) and lower volume of muscles involved, which could be insufficient to provoke significant metabolic acute adaptations.

The responses of the cardiovascular parameters (HR, SBP, and DBP) were in line with previous studies evaluating hemodynamic changes with the use of compression garments (*Lee et al., 2022*). It must be mentioned that, although more repetitions with greater loads and muscle activation were performed, nonsignificant differences were observed in almost all the cardiovascular parameters analyzed. Only a significantly greater DBP was obtained after performing the seated shoulder press wearing the elastomeric garment. Considering the compressive property of the elastomeric garment, this increase in DBP may be due to abdominal compression increasing mean arterial pressure and sympathetic nerve activity, which may increase cardiovascular responses (*Platts et al., 2009*; *Stenger et al., 2013*). However, this could be recognized as a normal physiological finding considering that DBP increases after exercise, including resistance training at maximal intensities, in healthy participants (*MacDougall et al., 1985*; *Wilborn et al., 2004*; *Lee et al., 2022*). This increase in DBP could be explained by the mechanical pressure of the muscles on the blood vessels and the pressor reflex generated during contraction (*Iglesias-Soler et al., 2015*; *Gjovaag et al., 2016*). Furthermore, the cardiovascular response to strength training is affected by several factors, including body position (*MacDougall et al., 1985*; *Wilborn et al., 2004*). More specifically, a previous study (*MacDougall et al., 1985*) reported further increases in DBP after performing resistance exercises (80, 90, 95, and 100%1RM) to muscle failure in an upright position compared to supine. Although we did not control this, another reason to consider is elevated intrathoracic and intraabdominal pressure during the Valsalva maneuver, which may increase DBP (*Wilborn et al., 2004*). Considering that our sample consisted of healthy young adults, the acute increase in DBP after exercise cannot be extrapolated to other participants, such as older adults. Therefore, new studies are warranted to monitor the effect of performing resistance training using elastomeric compressive garments on blood pressure responses in vulnerable populations.

### Mechanical variables

In our study, the use of the elastomeric sports garment for the push-ups allowed participants to perform a greater number of repetitions until the muscular failure, with significantly greater 1stMVP, PMVP, and AMPV. Similarly, participants performed significantly more repetitions with greater 1stMVP and PMVP in the seated shoulder press wearing the elastomeric garment compared to the placebo. These results may be due to the elastomeric garment allowing to overcome the sticking region in each repetition as previously mentioned, therefore, allowing greater movement speed with more kilograms.

In this sense, it could be suggested that the elastomeric garment, as elastic bands do, allows for a greater peak force and power (*Wallace, Winchester & McGuigan, 2006*; *Argus et al., 2011*; *Andersen et al., 2020*; *Babiloni-Lopez et al., 2022*). Furthermore, the increased time under tension (larger number of push-ups) increases glycolysis metabolism and could promote superior muscle adaptations by stimulating delayed muscle protein synthesis at 24–30 h of recovery (*Aboodarda et al., 2012*).

From a biomechanical point of view, the resistance provided by the elastomeric garment seems to match the resistance profile of both exercises. Additionally, attention should be paid to the potential assistance the elastomeric garment provides during the initial degrees of the concentric phase of these exercises. In this concern, the chest is opened (*i.e.*, scapulae retracted) at the lowest position of both exercises. Therefore, the front elastomers are stretched and assist the movement. Oppositely, the back elastomers are slack and, therefore, do not provide any resistance. This combination of factors would function as an elastic band attached to the ceiling to assist the movement, which allows a higher execution velocity (*Tran et al., 2012*). After that, when the concentric phase starts, the elastomers from the back begin to stretch and provide resistance until eight kilograms at the end of the range of motion. Finally, due to elastic properties, the maximum elongation of the elastomers at the end of the concentric phase may prestress and accelerate the next movement to be performed (*i.e.*, eccentric phase) (*Bartolini et al., 2011*).

## Psychological variables

Previous studies demonstrated strong inverse relationships between MPV and RPE ($r = -0.79$ to $-0.87$) (*Helms et al., 2017*). Controversially, participants from our study, although performing at greater MPV, perceived the use of the elastomeric garment for both exercises as less demanding (significantly lower RPE) compared to the placebo. This could be attributed to the properties of the elastomers and the decreased weight in the lower phases of both exercises. As for the RPE of the last repetition, no significant differences were observed between both garments in any of the exercises. The nonsignificant differences between the garments are interesting due to the participants using eight more kilograms, performing approximately three more push-ups, and 0.5 more repetitions of the seated shoulder press wearing the elastomeric garment compared to the placebo.

## Limitations and future research

It is crucial to acknowledge that the outcomes of this study are constrained by the specific independent and dependent variables examined and the sample size. Consequently, new variables could be analyzed, such as open or closed kinetic chains, exercises with displacement, and/or maximum joint mobility in all planes of motion simultaneously. Additionally, future studies should include larger samples to compare different physical fitness levels and training experience levels, group ages, and genders. Similarly, it would be interesting to analyze the movement velocity differentiating by specific phases of the range of motion, *e.g.*, (i) from the beginning to the first half of the range of movement, where the front elastomer is supposed to assist the movement; (ii) from the middle to the final point of the movement, where the rear elastomer resists. Performing the exercises with the

transition between eccentric and concentric phases at maximum velocity would provide new information on the potential effects of the elastomeric garment on the stretch-shortening cycle. Finally, considering that the resistance provided by the elastomeric garment is approximately eight kilograms, the relative load (*e.g.*, %1RM) used by each participant may slightly differ. This different relative load cannot be considered a bias in our present study as we applied a within-participant design. Future studies with a between-participant design should consider potential differences in the relative load.

## CONCLUSIONS

Wearing the elastomeric garment to perform both exercises allowed participants to obtain greater muscular activation, lifting velocity, and time under tension (more maximum number repetitions). Furthermore, the exercises performed with the elastomeric garment were not perceived as more strenuous, provoked less post-exercise blood lactate (in the push-ups), and showed no significant differences in HR and SBP compared to the placebo garment. This fact confirms that the use of the elastomeric garment can optimize the external load parameters while maintaining similar values of the internal load.

The present findings help to generate more practical, efficient, and healthy workouts based on assisting and resisting movement through the incorporation of elastomers in a sports garment. Traditional elastic band training is effective in generating positive neuromuscular adaptations, but they limit users for example, to single-plane movements, work on-site, and/or a determinate type of exercises. These limitations led us to believe that the training sessions could be improved if new tools were applied. Fortunately, the present sports garment, through elastomeric technology, may solve most of the limitations previously described for traditional elastic bands. Exercisers can now use the elastic variable resistance incorporated in their sportswear with no need to hold a handle or a complex setup.

## ACKNOWLEDGEMENTS

We thank the participants for their voluntary collaboration. We would also like to thank Menatechpro System® for providing us with elastomeric and placebo garments.

### Funding

The authors received no funding for this work.

### Competing Interests

This research has been developed under the advice and technical support contract signed between the University of Valencia (Spain) and Menatechpro System®. Dr. Juan C. Colado is the professor responsible for the University. The rest of the coauthors are members of the research group led by Dr. Juan C. Colado. Amador García-Ramos is an Academic Editor for PeerJ.

## Author Contributions

- Angel Saez-Berlanga conceived and designed the experiments, performed the experiments, analyzed the data, prepared figures and/or tables, authored or reviewed drafts of the article, and approved the final draft.
- Carlos Babiloni-Lopez conceived and designed the experiments, performed the experiments, authored or reviewed drafts of the article, and approved the final draft.
- Ana Ferri-Caruana conceived and designed the experiments, authored or reviewed drafts of the article, and approved the final draft.
- Pablo Jiménez-Martínez conceived and designed the experiments, performed the experiments, prepared figures and/or tables, authored or reviewed drafts of the article, and approved the final draft.
- Amador García-Ramos analyzed the data, prepared figures and/or tables, authored or reviewed drafts of the article, and approved the final draft.
- Jorge Flandez analyzed the data, authored or reviewed drafts of the article, and approved the final draft.
- Javier Gene-Morales conceived and designed the experiments, performed the experiments, analyzed the data, prepared figures and/or tables, authored or reviewed drafts of the article, and approved the final draft.
- Juan C. Colado conceived and designed the experiments, analyzed the data, prepared figures and/or tables, authored or reviewed drafts of the article, and approved the final draft.

## Human Ethics

The following information was supplied relating to ethical approvals (*i.e.*, approving body and any reference numbers):

The study protocol was approved by the Human Research Ethics Committee of the University of Valencia (H20190325095509).

## Data Availability

The raw data is available in the Supplemental Files.

## Supplemental Information

Supplemental information for this article can be found online at http://dx.doi.org/10.7717/peerj.17008#supplemental-information.

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
