# Peer review of "A new sports garment with elastomeric technology optimizes physiological, mechanical, and psychological acute responses to pushing upper-limb resistance exercises"

_PeerJ, doi:10.7717/peerj.17008_

## Round 0.1 · original submission · Major Revisions

Thank you for waiting

Please, attend reviewers´ comments.

Regards

Dr. Manuel Jiménez

**Language Note:** PeerJ staff have identified that the English language needs to be improved. When you prepare your next revision, please either (i) have a colleague who is proficient in English and familiar with the subject matter review your manuscript, or (ii) contact a professional editing service to review your manuscript. PeerJ can provide language editing services - you can contact us at copyediting@peerj.com for pricing (be sure to provide your manuscript number and title). – PeerJ Staff

Reviewer 1 ·

Basic reporting

The introduction, in general, is well constructed and directed, with adequate and easy to understand language.

The second paragraph of the introduction, lines 60-77 was a little disjointed. I suggest improving the cohesion between ideas. It was necessary to read it several times before I could understand that the objective of the paragraph was to summarize the different ways of evaluating training load (EMG, mechanical, metabolic and psychological components).

In lines 101-102 the authors state that the use of bands “limits their use to single-plane movements, work on-site, and/or other constraints”. I would like this statement to be substantiated. Both the use in practice and the analysis of the exercises that we see with the use of elastic bands allow us to realize that movements are worked on in different planes, including the combination of two or more planes in a single movement, in addition to being an easy piece of equipment. transport and adaptability to different locations.

I would like the correlation mentioned in lines 112-115, between the characteristics of the elastomer and the hypothesis created, to be better explained. I believe that, if elastomers can “resist or assist the movement”, the analyzed parameters could also respond in an ambiguous way and not just be optimized.

Experimental design

In lines 142-144 it was mentioned that some participants held an additional session to assess intersession reliability. Could this additional session not have interfered with the total training volume?

In line 178, the authors said that in the placebo situation the participants wore the same clothing but without the elastomer. Was it possible that the participants did not notice any difference between the outfits? Or did the clothes have, for example, the same weight, texture, resistance when wearing, etc.?

Was there the possibility of adjusting the size of the clothing or the elastomers (lines 182-184)?

The clothing with elastomers had the same final resistance of 8kg for all participants (lines 185-186). This load may represent a different intensity increase for each participant, depending on their 1RM. How was this analyzed by the authors?

The acronyms RMSIDELT, RMSABD e RMSTRI, not described earlier in the text (lines 328-329)

Validity of the findings

The manuscript results present a great degree of impact and news.
The results were well-structured and explained clearly and concisely. The discussion was able to address any uncertainties

Additional comments

It would be interesting to include an image of the equipment with the elastomers and their positioning on the volunteers.

Reviewer 2 ·

Basic reporting

no comment

Experimental design

no comment

Validity of the findings

no comment

Additional comments

The manuscript entitled “A new sports garment with elastomeric technology optimizes physiological, mechanical, and psychological acute responses to pushing upper-limb resistance exercises” evaluates the possibility of using a new sports garment for resistance training, from the multivariant perspectives. Despite its complexity, the manuscript is easy and interesting to read, the introduction hypotheses driven, the methods are well described, and the conclusions are well stated. I have made a few observations that I think could improve the manuscript from the perspective of readers.

Line 91.-93. Regarding this type of training device, a previous study confirmed that variable resistance training was the best, along with eccentric training, to enhance the benefits of hypertrophy, strength, and power (Suchomel et al., 2018). Variable resistance training (e.g., elastic bands) allows exercisers to employ a mechanical advantage ….

Comment
Most training studies of variable resistance, use elastic bands or chains as an addition ( ̴20% of total resistance) to free weight exercises, not only elastic bands alone. The suggestion is to include this explanation, for example in brackets.

Literature conclusions are not unambiguous about the superior effects of variable resistance over standard free weight training on all three performance-related qualities that you mentioned (e.g. some studies reported equal effect)

Line 96 - 97. Through the use of elastic bands to resist the movement, participants achieved superior strength gains and higher muscle activation levels (Aboodarda et al., 2013).

Comment
The study of „Aboodarda et al., 2013“ is not training but a biomechanical cross-sectional design - the conclusion that participants achieved superior strength gains is not appropriate.

Line 195-197 The specific relative and absolute intersession reliability values of each dependent variable can be found in Supplementary Tables S1 and S2.

Comment
I could not find the cited tables in the supplementary material.

Lines 227-229, To end with the familiarization session, participants performed one set to failure of each exercise (at 70%1RM for the seated shoulder press and bodyweight for the pushups) reporting the RPE at the first and last repetition of each set.

Comment
How much rest was allowed before and between each set to failure? Could this time condition the perceived effort?

Line 338, The test-retest relative reliability was assessed using the intraclass correlation coefficient (ICC)

Comment
What model of ICC was used?


Additional comments
From a practical perspective performing the same muscular endurance stimuli to failure separated with one day of rest (48 hours) is not the best choice for optimal recovery. Potentially, in the following testing session (testing for reliability) the fatigue level could be very high as well as interindividual differences. Potentially it could be a reason for the reliability decline.

What was the approximate duration of each session? Was there any prespecified breathing pattern during the exercise performance?

Reviewer 3 ·

Basic reporting

Regarding the basic structure, I consider that the paper presents the following weaknesses:
- The abstract does not present substantial information regarding the results.
- Introduction so long that it covers aspects related to training in general. However, the authors wrote little about the tested device.

Experimental design

The sample calculation is inadequate. No F statistic was used. Justify why you chose a large effect size. The sample calculation should be made considering the comparison between groups carried out in the study (comparison of means of dependent groups). The results found make it very clear that the sample is undersized. This needs to be discussed and included as a limitation of the study.
There is no information on procedures for normalizing EMG signals.
The description of signal analysis is very superficial. The authors report that EMG was synchronized with data from inertial sensors. Why were data for the concentric and eccentric phases not presented separately?
RMS values shown are averages. How many repetitions?

Validity of the findings

As everyone already knows, small samples can bias study findings. Furthermore, other problems need to be highlighted.

The authors report significant differences in the EMG activity of the PM, DA, and TB muscles. Furthermore, the authors declare that they normalized the EMG signal in the method section. That doesn't seem to have happened. Looking at the RMS values and the unit of measurement, it is clear that the authors analyzed the raw EMG data. This is a serious problem that compromises analyses on different days.

---

## Round 0.2 · accepted · Accept

Dear Co-authors:

Following the reviews received regarding your resubmission, I have the great pleasure of informing you that your work has been accepted for publication in PeerJ.

Thank you for your trust, and congratulations on your work.

Dr. Manuel Jimenez

Reviewer 1 ·

Basic reporting

Any doubts or concerns presented in this session were resolved satisfactorily.

Experimental design

Any doubts or concerns presented in this session were resolved satisfactorily.

Validity of the findings

Any doubts or concerns presented in this session were resolved satisfactorily.

Additional comments

After the corrections made, the manuscript is clearer and more cohesive. The necessary information was added allowing the entire text to be understood.

Reviewer 2 ·

Basic reporting

no comment

Experimental design

no comment

Validity of the findings

no comment

Additional comments

I am of the opinion that the authors significantly improved the manuscript, taking into account the suggestions and corrections of the reviewers.

Reviewer 3 ·

Basic reporting

No comment

Experimental design

No comment

Validity of the findings

No comment

Additional comments

No comment